# Ankle–Brachial Index Predicts Long-Term Renal Outcomes in Acute Stroke Patients

**DOI:** 10.3390/healthcare10050913

**Published:** 2022-05-13

**Authors:** Tsung-Lin Lee, Yu-Ming Chang, Chi-Hung Liu, Hui-Chen Su, Pi-Shan Sung, Sheng-Hsiang Lin, Chih-Hung Chen

**Affiliations:** 1Department of Neurology, National Cheng Kung University Hospital, College of Medicine, National Cheng Kung University, Tainan 704, Taiwan; c2481023@hotmail.com (T.-L.L.); cornworldmirror@hotmail.com (Y.-M.C.); shjmirage@gmail.com (H.-C.S.); lchih@mail.ncku.edu.tw (C.-H.C.); 2Stroke Center, Department of Neurology, Chang Gung Memorial Hospital, Linkou Medical Center and College of Medicine, Chang Gung University, Taoyuan 333, Taiwan; ivanliu001@cgmh.org.tw; 3Institute of Clinical Medicine, College of Medicine, National Cheng Kung University, Tainan 704, Taiwan; hsiang0922@gmail.com; 4Department of Public Health, College of Medicine, National Cheng Kung University, Tainan 704, Taiwan; 5Biostatistics Consulting Center, National Cheng Kung University Hospital, College of Medicine, National Cheng Kung University, Tainan 704, Taiwan

**Keywords:** ABI, baPWV, poststroke renal dysfunction, renal function trajectory after stroke

## Abstract

Renal dysfunction is common after stroke. We aimed to investigate the clinical predictability of the ankle–brachial index (ABI) and brachial-ankle pulse wave velocity (baPWV) on poststroke renal deterioration. A total of 956 consecutive participants with acute ischemic stroke between 1 July 2016, and 31 December 2017 were enrolled and a final of 637 patients were recruited for final analysis. By using the group-based trajectory model (GBTM), the patients’ renal function trajectories were grouped into the low, intermediate, and high categories (LC, IC, and HC). Significant deterioration in the slope was noted in the IC (*p* < 0.001) and LC (*p* = 0.002) groups but was nonsignificant in the HC (*p* = 0.998) group. Abnormal ABI (ABI ≤ 0.9) was independently related to LC (adjusted odds ratio: 2.40; 95% CI, 1.16–4.95; *p* = 0.019) and was also independently associated with increased risks of a ≥30% decline in eGFR (adjusted hazard ratio [aHR], 2.28; 95% CI, 1.29–4.05; *p* = 0.005), a doubling of serum creatinine (aHR, 3.60; 95% CI, 1.93–8.34; *p* < 0.001) and ESRD (HR, 3.28; 95% CI, 1.23–8.74; *p* = 0.018). However, baPWV did not have a significant relationship with any of the renal outcomes. Patients with a lower ABI during acute stroke should receive regular renal function tests and should be closely monitored to improve poststroke renal care.

## 1. Introduction

Renal dysfunction and chronic kidney disease (CKD) is a growing concern and its incidence continues to increase worldwide [1]. CKD is defined by an eGFR less than 60 mL/min/1.73 m^2^, proteinuria, or structural kidney disease. It is considered to pose a long-term burden on the health care system [2]. The disease burden of stroke is also increasing worldwide. According to the Global Burden of Disease, Injuries and Risk factors Study (GBD) 2017, stroke was the third-leading cause of death and disability combined [3]. Bidirectional influence between these two diseases may be noted. CKD is highly associated with cerebro-cardiovascular diseases (CVDs), but it is also a common disorder after ischemic stroke (IS), especially in those with small vessel disease and atheromatic IS [4]. Small vessel disease and atheromatic macroangiopathy could result in long-term damage to both the kidney and brain.

Overlap mechanisms may be noted in the disease spectra of stroke and CKD. For example, low-grade inflammation, a common feature of CKD, is critical in the pathogenesis of atherosclerosis [5]. Inflammation may predict worsening renal function in subjects without overt renal disease [6], cardiovascular complications in CKD patients [5] and may also be a risk factor for clinical stroke in addition to silent infarcts [7]. Other proposed causal mechanisms for promoting CKD patients having higher risks of CVDs include hyperhomocysteinemia, oxidative stress, and dysregulated levels of lipoproteins and calcium phosphorous metabolism [8]. However, systemic atherosclerotic burden may predict worsening renal disease. Patients with a history of myocardial infarction [9], the presence of cardiovascular diseases [10], or higher carotid intimal thickness [11], have been reported to have a higher risk of worsening renal function and the onset of end-stage renal disease (ESRD). Therefore, evaluating the burden of systemic atherosclerosis may potentially predict the change or worsening of renal function, including in stroke patients.

The ankle–brachial index (ABI) is a useful, noninvasive tool for predicting systemic atherosclerosis and the outcome of cardiovascular disease [12,13]. An ABI value lower than 0.9 indicates a high sensitivity and specificity for detecting lower extremity peripheral arterial disease (PAD) [14] and indicates the potential risk for future CVDs, renal disease, and hypertension [15]. The Global Atherothrombosis Assessment (AGATHA) study also showed that the frequency of an abnormal ABI is relatively higher in stroke patients [16]. The brachial–ankle pulse wave velocity (baPWV) is another well-established marker that is used to evaluate arterial stiffness. Several studies have demonstrated that a low ABI and high baPWV values are useful predictors for renal function decline [17,18] and incidence of cardiovascular events [19,20] in different patient cohorts, but the relationship between ABI and baPWV with poststroke renal outcomes has been far less investigated. Therefore, our study aims to investigate whether ABI and baPWV have the potential to become powerful predictors of poststroke renal deterioration or dysfunction by using a retrospective cohort study in a tertiary medical center.

## 2. Materials and Methods

### 2.1. Study Design and Dataset

This retrospective cohort study was conducted between 1 July 2016 and 31 December 2017, in National Cheng Kung University Hospital (NCKUH), a tertiary referral center in Taiwan. The study was approved by the Institutional Ethics Review Board of NCKUH (B-ER-107-186). The institutional ethics review board of NCKUH waived the need for informed consent. All methods were carried out in accordance with the relevant guidelines and regulations. We enrolled 956 participants aged 20–99 years with a history of acute ischemic stroke. The diagnosis was confirmed by magnetic resonance imaging (MRI). Prior to recruitment, detailed medical history was collected by the clinical physicians regarding their past medical history and functional status before the index stroke. After an acute stroke, the subjects’ medical history, previous medication use (i.e., antihypertensive or antidiabetic drugs), or demographic information was collected by trained medical staff. A self-reported history or medical record of clinical heart disease (including a history of documented heart failure and myocardial infarction) or hyperlipidemia was acquired. Further examinations, such as brain MRI, renal function, ABI, baPWV, body mass index (BMI), blood pressure, lipid profile, cardiac function, and diabetic profile, were checked soon after the index stroke. The modified Rankin Scale (mRS) was used to evaluate the functional status of each participant at discharge. A clinical follow-up schedule was arranged every 3 months after discharge. To assess the renal function trajectory after stroke, at least 2 sets of follow-ups of renal functions were needed at 6 months, 12 months, or >12 months after the index stroke. Patients who were lost to follow-up during scheduled clinical visits, refused blood serum tests, had missing baseline eGFR or creatinine data, had less than 2 sets of follow-up renal function data, or had a diagnosis of stroke mimics were excluded from this study.

### 2.2. Baseline Survey

The subjects’ body weight and height were each measured at admission and were used to calculate BMI as weight in kilograms divided by height in meters squared. BP measurements were obtained by trained and certified nursing staff members after >5 min of quiet rest. Hypertension was confirmed when systolic BP ≥ 140 mm Hg and/or diastolic BP ≥ 90 mm Hg and/or current use of antihypertensive medication.

For the renal function survey, we collected the patients’ serum creatinine results that were closest to the index date as the baseline value, which was usually in the acute stage of index stroke. The eGFR was calculated using the Chronic Kidney Disease Epidemiology Collaboration (CKD-EPI) equation [21]. The concentrations of blood sugar, HbA1C, total cholesterol, triglycerides, and LDL cholesterol were measured using standard laboratory methods. Diabetes was confirmed when fasting glucose ≥ 7.0 μmol/L (>126 mg/dL), or a random glucose ≥ 11.1 mmol/L (>200 mg/dL), and/or use of insulin or other anti-diabetic medication [22]. All laboratory analyses were conducted at the NCKUH laboratory with strict quality control.

After stroke was confirmed, the patients’ ABI or baPWV was measured during hospitalization. The ABI was defined as the ratio of systolic blood pressure in the posterior tibial artery and/or the dorsalis pedis artery to systolic blood pressure in the brachial artery [23]. Patients were asked to lie in the supine position for 5 min before ABI measurement. We used an automated, validated ABI-form device (vs. −1000; Fukuda Denshi Co. Ltd. Tokyo, Japan), which simultaneously measures blood pressure in both the arms and the ankles by using the oscillometric method for the measurement of ABI. The lowest ABI value or the average value calculated from the right and left ABI was taken as the patient’s ABI value and used in the analysis. The ABI was measured once in each patient. In each patient, we defined an ABI that was less than 0.9 as the low ABI group, an ABI that ranged from 0.9 to 1.0 as the low to normal group, and an ABI that was higher than 1.0 as the normal group.

The baPWV values were measured using an ABI-form device (vs. −1000; Fukuda Denshi Co. Ltd. Tokyo, Japan), which simultaneously records pulse volume waveforms of the brachial and posterior tibial arteries, as well as an automated oscillometric method [24]. The baPWV was measured in the supine position after 5 min of rest at room temperature, which was approximately 25 °C. The transmission time (Tba) was measured as the time interval between the initial increase in the brachial and ankle waveforms. The transmission distance was measured from the brachium to the ankle according to body height. The distance (Lb-La) between the path length from the suprasternal notch to the brachium (Lb) and from the suprasternal notch to the ankle (La) was calculated based on the patient’s height, and the time delay from the ascending point of the brachial waveform and each ankle waveform (DTba) was obtained. The baPWV was defined as the pulse wave propagation distance (Lb-La) divided by the propagation time (DTba) and expressed in m/s [25]. The average systolic and diastolic BP of the bilateral arms was used for analysis. The baPWV was measured once in each patient.

### 2.3. Outcome Definition and Statistical Analysis

The participants in this study were divided into the following categories according to ABI values: ABI ≤ 0.90, ABI 0.91~0.99, and ABI ≥ 1.00 and according to baPWV values: baPWV < 1.4 and baPWV ≥ 1.4 m/s. We applied two different models for the renal outcome analysis. The first model is the group-based trajectory model (GBTM) [26]. GBTM will classify patients’ poststroke renal function into different trajectories, and the Bayesian information criterion will be used as the criterion for model selection. Then, we used multinomial logistic regression analysis to determine the factors that impact the trajectory of renal function after stroke, especially ABI, baPWV, and other potential clinical factors. The second model is a survival analysis model [27] that measures the time-to-event outcome. In the second part, the renal outcomes were defined as follows: (1) ≥30% decline in eGFR (2) doubling of the serum creatinine level to represent earlier and (3) ESRD. A ≥30% decline in eGFR and a doubling of serum creatinine were defined as changes in values with respect to the baseline at any time point during the follow-up, but the decline may need to persist for more than 3 months. A lesser reduction (≥30%) of eGFR is a novel, validated definition for early renal function decline and has been acknowledged as an endpoint of predicting end-stage renal disease in clinical research [28]. Doubling of serum creatinine is often used as a marker for worsening kidney function in nephrology trials and also predicts the development of ESRD [29,30]. It is an accepted endpoint by the Food and Drug Administration for clinical trials of renal dysfunction. Kidney failure or ESRD is defined as an eGFR that is lower than 15 mL/min per 1.73 m^2^, history of a previous kidney transplant or undergoing long-term dialysis [28]. A Cox proportional hazards model was used to estimate the hazard ratios (HRs) and 95% confidence intervals (CIs) for developing renal outcomes in each ABI and baPWV category, with ABI 1.00~1.40 and baPWV < 1.4 acting as each reference category.

These two models incorporated the following variables as covariates: age (years as a continuous variable), sex (male or female), BMI (kg/m^2^ as a continuous variable), hyperlipidemia (present or absent), heart disease (present or absent), antihypertensive drug use (present or absent), antidiabetic drug use (present or absent), and mRS at discharge (≥2 or <2). All statistical procedures were performed with the statistical software package SAS for Windows (version 9.2; SAS Institute Inc., Cary, NC, USA). Data are expressed as the mean ± standard deviation or as a percentage. A *p* value of 0.05 was considered statistically significant.

## 3. Results

In this study, a total of 956 patients’ data were retrospectively collected, and a total of 637 patients were enrolled in the analysis according to the above exclusion criteria (Figure 1). The mean follow-up duration was 26.9 ± 10.2 months and the average number of follow-ups was 9 ± 3.4 visits. In the baseline characteristics of the study population, 389 subjects were men (61.1%), the mean age was 68.3 ± 13.3 years, the mean BMI was 24.2 ± 4.1 kg/m^2^, 59.0% of the patients had hyperlipidemia, 17.7% of the patients had heart disease, 32.7% of the patients had diabetes mellitus, and 67.8% of the patients were hypertensive. The prevalence of an abnormal ABI was 17.7%, the prevalence of high baPWV was 92.7%, and the mean eGFR was 72.0 ± 22.0 mL/min/1.73 m^2^. Patients with an ABI ≤ 0.9 or a baPWV ≥ 1.4 tended to be older and have a higher prevalence of hypertension and renal dysfunction (Table 1).

Among the 637 patients, the GBTM model classified the trajectories of poststroke renal function into three groups: low category (LC), intermediate category (IC), and high category (HC) with different baseline data of mean eGFR 84.22 ± 10.00 mL/min/1.73 m^2^, 60.41 ± 15.13 mL/min/1.73 m^2^, and 25.91 ± 13.80 mL/min/1.73 m^2^ among the three groups, respectively (Figure 2). Significant deterioration in the slope was noted in the IC (*p* < 0.001) and LC (*p* = 0.002) groups but was nonsignificant in the HC group (*p* = 0.998) (Figure 2). By using the HC group as the reference group, the multinomial logistic regression revealed that an abnormal ABI (ABI ≤ 0.90) (adjusted odds ratio, aOR: 2.40, 95% CI 1.16–4.95, *p* = 0.019), but not high baPWV, independently predicted that a patient’s poststroke renal trajectory would be classified into the LC group. Older age was also a predictive factor for poststroke renal function in the IC group and LC group, but sex, BMI, heart disease, hyperlipidemia, hypertension, diabetes mellitus, and mRS at discharge did not show a significant impact on the prediction of poststroke renal function trajectory (Table 2).

In the second statistical model, the incidence rates of a ≥30% decline in eGFR, a doubling of serum creatinine, and ESRD in patients with an abnormal ABI were 24.8%, 13.3%, and 6.5%, respectively, and in patients with a higher baPWV, were 16.1%, 5.7%, and 4.1%, respectively. Older age and an abnormal ABI were independently associated with increased risks of a ≥30% decline in eGFR during follow-up (age: aHR: 1.03; 95% CI, 1.01–1.06; *p* = 0.002; abnormal ABI: aHR: 1.90; 95% CI, 1.09–3.34; *p* = 0.025) (Table 3). DM may be associated with an increased risk of a ≥30% eGFR decline (aHR: 1.61; 95% CI, 1.00–2.60; *p* = 0.05), but other factors, such as sex, BMI, higher baPWV, heart disease, hyperlipidemia, hypertension, smoking, and poor discharge mRS (≥2), had no independent negative impact on poststroke eGFR decline. An abnormal ABI also independently increased the risk of a doubling in serum creatinine (aHR: 3.60; 95% CI, 1.64–7.91; *p* = 0.001) and ESRD (aHR: 3.28; 95% CI, 1.23–8.74; *p* = 0.018) during follow-up (Table 3). The association between an abnormal ABI and poor renal outcome, including the risk of serum creatinine doubling (adjusted CRR: 3.15; 95% CI: 1.42–6.96) and ESRD (adjusted CRR: 2.48; 95% CI: 1.00–6.17) was persistent after adjusting for death (Appendix A).

We further analyzed the risk of different renal outcomes in different ABI categories (ABI ≤ 0.7, 0.7 < ABI ≤ 0.9, 0.9 < ABI ≤ 1.1, ABI > 1.1), where a significant increasing trend of a higher incidence rate of a ≥30% decline in eGFR was found while ABI values decreased (*p* = 0.048), and the highest incidence rate (26.83%) was in the ABI ≤ 0.7 group. The significantly increasing incidence rate of a doubling in serum creatinine was also found with a similar trend (*p* = 0.004), but the trend was not obvious in the occurrence of ESRD during follow-up (Table 4).

## 4. Discussion

Our study found that a significant decline in the poststroke renal function trajectory would occur in patients with a reduced eGFR during the index stroke (an eGFR below the definition of stage II CKD). In addition, patients with a lower ABI, rather than abnormal baPWV, were highly and significantly associated with an increased risk of being in poor poststroke renal function trajectory and in future have a poor poststroke renal outcome.

In our cohort of stroke patients, the CKD prevalence rate was 26%, which was higher than the global population (9.1% in 2017) but similar to other stroke populations (varying from 20% to 35%) [31,32]. Stroke is well known to be associated with risk factors that promote atherosclerosis, such as smoking, diabetes, hypercholesterolemia, and arterial hypertension [33]. These factors are also associated with CKD [34]. A prospective analysis in Poland that involved 352 poststroke survivors showed a higher incidence of renal dysfunction in poststroke patients (especially those with lacunar or atheromatic subtypes) [4]. Additionally, a national nationwide population cohort in the Taiwan National Health Insurance program showed that both male and female stroke patients in Taiwan experienced an increased risk of ESRD [35]. Our study demonstrated that in poststroke patients, the renal function trajectory may be different in patients with differing renal function in the acute stage, with a significant trend of renal deterioration in the IC (mean initial eGFR: 60.41 ± 15.13 mL/min/1.73 m^2^) and LC groups (mean initial eGFR: 25.91 mL/min/1.73 m^2^). Therefore, those patients with an eGFR below the definition of stage II CKD at the acute stage require close and regular follow-ups of renal functions to improve poststroke care.

Our findings in the GBTM model revealed that old age and a low ABI, but not a high baPWV, independently predicted that a patient would be classified into the LC group during follow-up. In addition, we noted that the abnormal ABI group was also significantly associated with a nearly 2-fold increased risk of a ≥30% decline in eGFR and a nearly 3.5-fold increased risk of a doubling in serum creatinine and developing ESRD, even after adjusting for vascular comorbidities and the risk of death. In addition, we found that a lower ABI potentially indicated a higher incidence rate of a ≥30% decline in eGFR and a doubling in serum creatinine, similar to a dose-dependent manner (Table 4). However, baPWV did not have a statistically significant relationship with any of the renal outcomes. Although there was a relatively higher prevalence of an abnormal ABI in the preexisting CKD population (28% vs. 12.3% in CKD and non-CKD populations in a multicenter prospective project [36]), ABI has also been reported as a prognostic marker for renal deterioration in various patient populations, including healthy populations (Western or Asian populations), CKD populations [12], and our poststroke cohort. Lower ABI values indicate the presence and severity of flow-limiting atherosclerotic stenosis in the lower extremity arteries. It is a common risk factor for CVD and CKD due to the vascular burden of systemic atherosclerosis [37]. The relationship between low ABI and CKD may be explained by the mechanism involving generalized atherosclerosis-related intrarenal arteriolar hyalinization and glomerulosclerosis [38]. Therefore, a lower ABI indicates the possibility of more severe generalized atherosclerosis and more severe preexisting damage in renal vasculature and is thus a predictor of future renal decline in various patient populations, including stroke. However, baPWV is another well-established marker used to evaluate arterial stiffness. Some studies have shown that arterial stiffness independently predicts the progression of CKD [17], but controversial reports regarding the relationship between baPWV and CKD exist. In those studies, carotid–femoral pulse wave velocity (cfPWV) was measured and considered a more accurate measurement for arterial stiffness, and it directly reflects the aortic PWV [39]. Therefore, baPWV may not be as accurate as cfPWV because baPWV reflects the status of both the central and peripheral arteries, and cfPWV is widely related to the aorta, which may represent a greater association with large artery stiffness and subclinical target organ damage (TOD). Future studies regarding the potential predictability of cfPWV and renal function change may be warranted.

There are some limitations in our study. We monitored the eGFR and creatinine level as indicators of renal decline, and no further data on albuminuria or urine albumin to creatinine ratio (ACR) were specified in this study. However, due to incomplete clinical information, we did not include some known risk factors for renal diseases, such as patients’ lifestyle (e.g., medication use, alcoholism, smoking) and socioeconomic status. In addition, the study did not comprise the detailed data regarding the control status of comorbidities, including dynamic blood pressure, HbA1C/serum glucose level or low-density lipoprotein (LDL) level during follow-up. These variables could not be adjusted in the models. Third, the detailed drug prescriptions during follow-up, including any exposure to nephrotoxic drugs or renoprotective agents, such as angiotensin receptor antagonists or newer classes of oral antihyperglycemic agents, were not available for the present analysis. Therefore, the drug effect on renal function trajectory or renal outcomes could not be assessed in this study. Then, for a more accurate evaluation of arterial stiffness, other measurements, such as cfPWV, radial–dorsalis pedis PWV, and/or the augmentation index, may be considered. Finally, there was no comparison of the etiology of ischemic stroke in this study, which may also be an important indicator for generalized atherosclerosis. Previous studies have shown that the glomerular filtration rate (GFR) may be correlated with cerebral small-vessel disease. Therefore, further studies should be conducted to explore the underlying etiology and pathogenesis, which may be helpful for individualized intervention strategies for patients who are experiencing stroke.

## 5. Conclusions

Our findings illustrated that deterioration of poststroke renal function trajectory may be noted in those patients with eGFR below the definition of stage II CKD at the acute stage of stroke. ABI can potentially be an independent predictor of poor poststroke renal outcome. Patients with a lower ABI and who are older in age, without regard to baPWV values, may consider receiving regular renal function monitoring to improve poststroke care.

## Figures and Tables

**Figure 1 healthcare-10-00913-f001:**
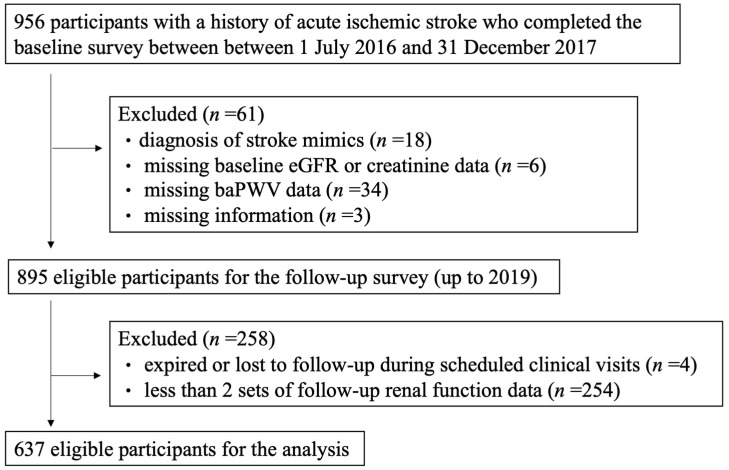
The flowchart of patient enrollment in this study.

**Figure 2 healthcare-10-00913-f002:**
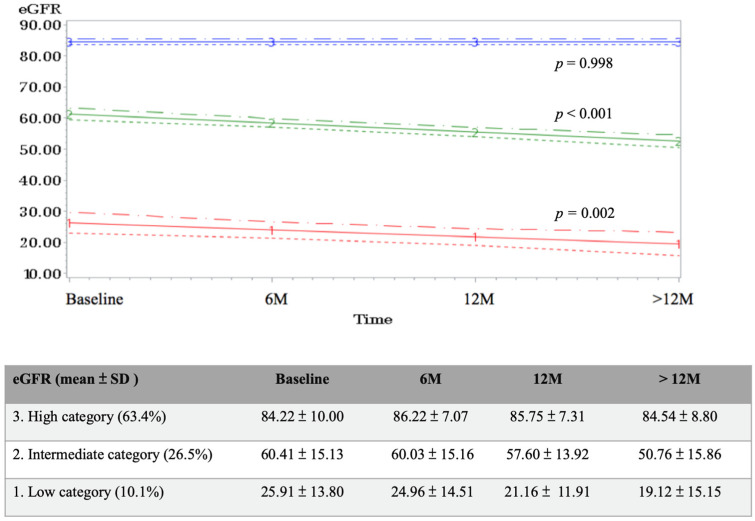
The poststroke renal function trajectory during follow-up. 6 M: 6 months after discharge; 12 M: 12 months after discharge; >12 M: more than 12 months after discharge.

**Table 1 healthcare-10-00913-t001:** Baseline characteristics and comorbidities of the enrolled subjects with acute ischemic stroke.

		Enrolled Subjects (*n* = 637) ^a^					
		ABI > 0.9	ABI ≤ 0.9	*p*	baPWV < 14	baPWV ≥ 14	*p*
*n*		524	113		44	559	
Age (mean ± SD)	[years]	66.7 ± 12.8	75.4 ± 13.2	<0.001	56.4 ± 19.5	68.9 ± 12.3	<0.001
Male	*n* [%]	323 (61.6)	66 (58.4)	0.594	28 (63.64)	345 (61.72)	0.927
BMI (mean ± SD)	[kg/m^2^]	24.5 ± 4.0	22.9 ± 4.3	<0.001	24.0 ± 4.4	24.3 ± 4.1	0.977
Hyperlipidemia	*n* [%]	308 (58.8)	68 (60.2)	0.866	24 (54.6)	334 (59.8)	0.605
Heart disease	*n* [%]	36 (6.9)	13 (11.5)	0.138	8 (18.2)	92 (16.5)	0.932
Hypertension	*n* [%]	340 (65.4)	88 (77.9)	0.014	17 (38.6)	389 (70.1)	<0.001
Diabetes Mellitus	*n* [%]	168 (32.31)	39 (34.5)	0.732	7 (15.9)	189 (34.1)	0.021
ABI ≤ 0.9	*n* [%]	-	-	-	10 (22.7)	92 (16.5)	0.390
baPWV ≥ 1.4	*n* [%]	467 (93.2)	92 (90.2)	0.390	-	-	-
eGFR (mean ± SD)	[mL/min/1.73 m^2^]	73.7 ± 21.5	64.3 ± 23.2	<0.001	76.1 ± 19.6	72.1 ± 22.0	0.205

^a^ A total of 34 missing data in baPWV group. ABI: ankle–brachial index; BMI: body mass index; baPWV: brachial–ankle pulse wave velocity.

**Table 2 healthcare-10-00913-t002:** Predictors of poor poststroke renal function trajectory by multinomial logistic regression (HC as a reference group).

	IC vs. HC				LC vs. HC			
Variables	Crude OR (95% CI)	*p* Value	Adjusted OR (95% CI)	*p* Value	Crude OR (95% CI)	*p* Value	Adjusted OR (95% CI)	*p* Value
Age	1.05 (1.03–1.06)	<0.001	1.05 (1.03–1.07)	<0.001 *	1.04 (1.02–1.06)	0.001	1.03 (1.01–1.06)	0.017 *
Male	0.95 (0.66–1.37)	0.772			1.13 (0.65–1.95)	0.672		
BMI	1.01 (0.96–1.05)	0.774			0.96 (0.89–1.02)	0.191		
Borderline ABI (0.91~0.99)	1.17 (0.69–1.98)	0.560	0.79 (0.44–1.41)	0.421	2.12 (1.05–4.29)	0.037	1.55 (0.71–3.40)	0.274
Abnormal ABI (≤0.90)	1.94 (1.21–3.08)	0.006	1.25 (0.73–2.15)	0.413	3.15 (1.67–5.96)	<0.001	2.40 (1.16–4.95)	0.019 *
baPWV ≥ 1.4	1.31 (0.63–2.75)	0.469			1.60 (0.47–5.42)	0.447		
Hyperlipidemia	0.90 (0.63–1.30)	0.585			0.92 (0.54–1.57)	0.757		
Heart disease	1.46 (0.75–2.85)	0.263	1.00 (0.45–2.23)	0.995	2.47 (1.10–5.58)	0.029	1.24 (0.44–3.51)	0.690
Smoking	0.51 (0.25–1.05)	0.067			0.36 (0.08–1.61)	0.183		
Hypertension	1.30 (0.88–1.91)	0.189			1.77 (0.96–3.26)	0.069		
Diabetes Mellitus	0.92(0.63–1.36)	0.683			1.43 (0.83–2.46)	0.193		
Poor discharge mRS (≥2)	1.73 (1.14–2.60)	0.009	1.26 (0.81–1.97)	0.303	1.63 (0.89–2.99)	0.113	1.09 (0.57–2.08)	0.802

* ABI: The minimal value of left ABI or right ABI; baPWV: using the maximal value of left baPWV or right baPWV. HC: High category; IC: Intermediate category; LC: Low category; BMI: body mass index; ABI: ankle–brachial index; baPWV: brachial–ankle pulse wave velocity; mRS: modified Rankin Scale; OR: Odds ratio.

**Table 3 healthcare-10-00913-t003:** Predictors of different renal outcomes during follow-up by the Cox proportional hazards model.

	A 30% Decline in eGFR		Serum Creatinine Doubling		ESRD	
Variables	Crude HR (95% CI)	Adjusted HR (95% CI)	Crude HR (95% CI)	Adjusted HR (95% CI)	Crude HR (95% CI)	Adjusted HR (95% CI)
Age	1.04 (1.02–1.06)	1.03 (1.01–1.06) *	1.04 (1.01–1.07)	1.02 (0.99–1.05)	1.05 (1.01–1.09)	1.04 (1.00–1.08) *
Male	0.77 (0.50–1.18)	-	0.99 (0.51–1.91)	-	1.36 (0.56–3.28)	-
BMI	0.99 (0.93–1.04)	-	0.94 (0.87–1.03)	-	0.97 (0.88–1.08)	-
Borderline ABI (0.91~0.99)	1.33 (0.72–2.46)	1.15 (0.60–2.22)	2.09 (0.86–5.09)	1.98 (0.78–4.98)	1.06 (0.24–4.68)	0.91 (0.21–4.04)
Abnormal ABI (≤0.90)	2.24 (1.35–3.71)	1.90 (1.09–3.34) *	4.35 (2.15–8.80)	3.60 (1.64–7.91) *	4.04 (1.56–10.45)	3.28 (1.23–8.74) *
baPWV ≥ 1.4	0.99 (0.40–2.46)	-	0.54 (0.19–1.53)	-	0.59 (0.14–2.56)	-
Hyperlipidemia	0.89 (0.60–1.31)	-	1.08 (0.55–2.11)	-	1.17 (0.50–2.73)	-
Heart disease	0.95 (0.47–1.92)	-	0.65 (0.20–2.14)	-	1.03 (0.30–3.52)	-
Smoking	0.84 (0.37–1.94)	-	1.01 (0.33–3.09)	-	0.99 (0.27–3.70)	-
Hypertension	1.47 (0.91–2.39)	-	1.65 (0.80–3.42)	-	1.36 (0.56–3.30)	-
Diabetes Mellitus	1.66 (1.08–2.55)	1.61 (1.00–2.60) *	1.51 (0.79–2.88)	-	1.40 (0.62–3.16)	-
Poor discharge mRS (≥2)	2.02 (1.18–3.44)	1.45 (0.84–2.52)	2.57 (1.12–5.90)	1.74 (0.73–4.12)	1.21 (0.52–2.84)	-

* ABI: The minimal value of left ABI or right ABI; baPWV: using the maximal value of left baPWV or right baPWV. BMI: body mass index; ABI: ankle–brachial index; baPWV: brachial–ankle pulse wave velocity; HR: hazard ratio; mRS: modified Rankin scale.

**Table 4 healthcare-10-00913-t004:** Predictors of different renal outcomes during follow-up by different ABI categories.

**(1) eGFR Decline > 30%**					
	ABI group				*p* Value
	ABI > 1.1 (*n* = 172)	0.9 < ABI ≤ 1.1 (*n* = 351)	0.7 < ABI ≤ 0.9 (*n* = 72)	ABI ≤ 0.7 (*n* = 41)	
	*n* (%)	*n* (%)	*n* (%)	*n* (%)	
No	150 (87.21)	297 (84.62)	55 (76.39)	30 (73.17)	0.048
Yes	22 (12.79)	54 (15.38)	17 (23.61)	11 (26.83)	
**(2) Serum** **Creatinine** **Doubling**					
	ABI group				*p* Value
	ABI > 1.1 (*n* = 172)	0.9 < ABI ≤ 1.1 (*n* = 351)	0.7 < ABI ≤ 0.9 (*n* = 72)	ABI ≤ 0.7 (*n* = 41)	
	*n* (%)	*n* (%)	*n* (%)	*n* (%)	
No	165 (95.93)	335 (95.44)	64 (88.89)	34 (82.93)	0.004
Yes	7 (4.07)	16 (4.56)	8 (11.11)	7 (17.07)	
**(3) The Occurrence of ESRD**					
	ABI group				*p* Value
	ABI > 1.1 (*n* = 171)	0.9 < ABI ≤ 1.1 (*n* = 339)	0.7 < ABI ≤ 0.9 (*n* = 68)	ABI ≤ 0.7 (*n* = 39)	
	*n* (%)	*n* (%)	*n* (%)	*n* (%)	
No	164 (95.91)	329 (97.05)	64 (94.12)	36 (92.31)	0.268
Yes	7 (4.09)	10 (2.95)	4 (5.88)	3 (7.69)	

## Data Availability

The data will be available upon request.

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
