# Peer review of "Ankle–Brachial Index Predicts Long-Term Renal Outcomes in Acute Stroke Patients"

_healthcare, 2022, doi:10.3390/healthcare10050913_

Round 1

Reviewer 1 Report

I received a manuscript for review entitled: Ankle-Brachial Index Predicts Long-Term Renal Outcomes in 2 Acute Stroke Patients. The manuscript is well written and touches on interesting topics.
As a reviewer, I have a few comments:
-The introduction is modest. There is a lack of information between stroke and inflammatory disease.
- The age range of 20-99 years. Did the size of this range affect the test results? Age should be a factor of adjustment
- The study of the groups differed significantly in age.
- There is no explanation of the abbreviations under the table, which makes it difficult to interpret
- Tables 3,4,5 are unreadable. There is a lot of unnecessary data in them. Maybe make one table out of them?
- In my opinion, the conclusions are too optimistic for the presented results.

Author Response

Dear Reviewer:

We thanks for the reviewer’s precious comments and suggestions. We revised our manuscript according to reviewer’s comments and suggestions pointedly.

Thank you again.

Reviewer 1

I received a manuscript for review entitled: Ankle-Brachial Index Predicts Long-Term Renal Outcomes in Acute Stroke Patients. The manuscript is well written and touches on interesting topics.

As a reviewer, I have a few comments:

  1. The introduction is modest. There is a lack of information between stroke and inflammatory disease.

Reply:

Thank you for the comments. We added information regarding the relationship between stroke, inflammation and CKD. Thank you for your helpfulcomments.

“ Page 1, Line 43 : Overlap mechanisms may be noted in the disease spectra of stroke and CKD. For example, low-grade inflammation, a common feature of CKD, is critical in the pathogenesis of atherosclerosis [5]. Inflammation may predict worsening renal function in subjects without overt renal disease [6],  cardiovascular complications in CKD patients [5] and may also be a risk factor for clinical stroke in addition to silent infarcts [7]. Other proposed causal mechanisms for promoting CKD patients having higher risks of CVDs include hyperhomocysteinemia, oxidative stress, and dysregulated levels of lipoproteins and calcium phosphorous metabolism [8]. However, systemic atherosclerotic burden may predict worsening renal disease. Patients with a history of myocardial infarction [9], the presence of cardiovascular diseases [10], or higher carotid intimal thickness [11], have been reported to have a higher risk of worsening renal function and the onset of end-stage renal disease (ESRD). Therefore, evaluating the burden of systemic atherosclerosis may potentially predict the change or worsening of renal function, including in stroke patients. “

  1. The age range of 20-99 years. Did the size of this range affect the test results? Age should be a factor of adjustment

Reply:

Thank you for the comments. Here, the meaning of 20-99 years indicated that we enrolled nearly all of our ischemic stroke patients, except for the patients with extremely young or extremely old age. We recognized that age would be an important factor influencing renal outcome. Therefore, in Model 1 (GBTM) and Model 2 (survival analysis model), age is listed as a covariate for adjustment [On page 4, line 164: These two models incorporated the following variables as covariates: age (years as a continuous variable), sex (male or female), BMI (kg/m2 as a continuous variable), hyperlipidemia (present or absent), heart disease (present or absent), antihypertensive drug use (present or absent), antidiabetic drug use (present or absent), and mRS at discharge (≥2 or <2). Since the effect of age was adjusted, the result may not be influenced by this enrollment criterion. Thank you for your helpful comments.

  1. The study of the groups differed significantly in age.

Reply:

Thank you for the comments. As the reviewer’s comment, we also noted that patients with ABI > 0.9 and baPWV ³ 14 were older than those with ABI < 0.9 and baPWV < 14. Therefore, age is an important covariate to investigate the independent predictability of ABI and baPWV on poststroke renal outcome. Therefore, we adopted age as a covariate for adjustment in Model 1 and Model 2. We also found that age was also an independent predictor of poor renal outcome, including an eGFR decline of more than 30% and the occurrence of ESRD (revised Table 3). Thank you for your helpful comments.

  1. There is no explanation of the abbreviations under the table, which makes it difficult to interpret

Reply:

Thank you for the comments. We added abbreviations to the table. Thank you for your helpful comments.

  1. Tables 3,4,5 are unreadable. There is a lot of unnecessary data in them. Maybe make one table out of them?

Reply:

Thank you for the comments. We summarized Tables 3-5 to Table 3. We moved some of the information to the supplementary data (competing risk ratio, CRR, for each renal outcomes). Thank you for your helpful comments.

  1. In my opinion, the conclusions are too optimistic for the presented results.

Reply:

Thank you for the comments. We adjusted the conclusion and hope to make the conclusion more practical. Thank you for your helpful comments.

“Page 10, line 385: ABI can potentially be an independent predictor of poor poststroke renal outcome. Patients with a lower ABI and who are older in age, without regard to baPWV values, may consider receiving regular renal function monitoring to improve poststroke care.”

Reviewer 2 Report

Proposed paper is interesting and well written. However, some revisions are needed before it can be accepted for publication:

  • In the introduction a better explanation of the background that lead to the study is needed. So please better explain why atherosclerotic measurements (ABI and baPWV) could be associated to renal outcomes in stroke patients.
  • How was heart disease defined?
  • More than dichotomous covariates, continuous one should be used. Are BP, glucose and LDL data available? if yes please substitute the continuous variables to their relative hypertension, diabetes, hyperlipidemia variable. If not available please state this as a limitation of the paper.
  • Please indicate in the ABS that 637 and not 956 patients were included in the present analysis.
  • What about therapies? are ARB/ACE-I more frequent in the group in which no differences in the renal outcomes are observed? are diuretic more frequently took by the opposite group? nephroprotective vs nephrotoxic therapies should be indicated and putted into the model. If not available please state this as a strong limitation.
  • The number of subjects at each follow-up visit, how follow-up data were taken it is not cleat nor is stated anywhere in the paper. Please clarify.
  • More than complicated logistic regression analysis continuous one are very important. Dichotomization always means lost of information. Please provide also data on correlation and linear regression model of GFR with ABI and baPWV.
  • Regarding ABI one important paper related to its use in cardiovascular prevention could be cited (10.1016/j.atherosclerosis.2020.11.004.)

Author Response

Dear Reviewer:

We thanks for the reviewer’s precious comments and suggestions. We revised our manuscript according to reviewer’s comments and suggestions pointedly.

Thank you again.

Reviewer 2

Proposed paper is interesting and well written. However, some revisions are needed before it can be accepted for publication:

  1. In the introduction a better explanation of the background that lead to the study is needed. So please better explain why atherosclerotic measurements (ABI and baPWV) could be associated to renal outcomes in stroke patients.

Reply:

Thank you for the comments. We added information regarding why atherosclerotic measurements may be associated with renal outcomes in stroke patients. Thank you for your helpful comments.

“Page 1, Line 43: Overlap mechanisms may be noted in the disease spectra of stroke and CKD. For example, low-grade inflammation, a common feature of CKD, is critical in the pathogenesis of atherosclerosis [5]. Inflammation may predict worsening renal function in subjects without overt renal disease [6], cardiovascular complications in CKD patients [5] and may also be a risk factor for clinical stroke in addition to silent infarcts [7]. Other proposed causal mechanisms for promoting CKD patients having higher risks of CVDs include hyperhomocysteinemia, oxidative stress, and dysregulated levels of lipoproteins and calcium phosphorous metabolism [8]. However, systemic atherosclerotic burden may predict worsening renal disease. Patients with a history of myocardial infarction [9], the presence of cardiovascular diseases [10], or higher carotid intimal thickness [11], have been reported to have a higher risk of worsening renal function and the onset of end-stage renal disease (ESRD). Therefore, evaluating the burden of systemic atherosclerosis may potentially predict the change or worsening of renal function, including in stroke patients.”

  1. How was heart disease defined?

Reply:

Thank you for the comments. We added information regarding the definition of heart disease on page 2, line 101 (including the history of documented heart failure and myocardial infarction). Thank you for your helpful comments.

  1. More than dichotomous covariates, continuous one should be used. Are BP, glucose and LDL data available? if yes please substitute the continuous variables to their relative hypertension, diabetes, hyperlipidemia variable. If not available please state this as a limitation of the paper.

Reply:

Thank you for the comments. Due to the original study design, we did not registered the dynamic laboratory data during follow-up. Therefore, we could not obtain detailed biochemical data for adjustment. We added this information as one of our study limitations. Thank you for your helpful comments.

“ Page 9, line 358: In addition, the study did not comprise the detailed data regarding the control status of comorbidities, including dynamic blood pressure, HbA1C/serum glucose level or low-density lipoprotein (LDL) level during follow-up. These variables could not be adjusted in the models.”

  1. Please indicate in the ABS that 637 and not 956 patients were included in the present analysis.

Reply:

Thank you for the comments. We revised this part in the abstract to make the description clearer. Thank you for your helpful comments.

  1. What about therapies? are ARB/ACE-I more frequent in the group in which no differences in the renal outcomes are observed? are diuretic more frequently took by the opposite group? nephroprotective vs nephrotoxic therapies should be indicated and putted into the model. If not available please state this as a strong limitation.

Reply:

Thank you for the comments. Due to the original study design, we did not include detailed drug prescription data in this study. Therefore, we could not assess the drug effect, including the exposure of nephrotoxic drugs or renoprotective agents, such as angiotensin-receptor antagonists or newer classes of oral antihyperglycemic agents, on renal function trajectory or renal outcome in this study. Therefore, we added this part as one of our major study limitations. Thank youfor your helpful comments.

“ Page 9, line 362: Third, the detailed drug prescriptions during follow-up, including any exposure to nephrotoxic drugs or renoprotective agents, such as angiotensin-receptor antagonists or newer classes of oral antihyperglycemic agents, were not available for the present analysis. Therefore, the drug effect on renal function trajectory or renal outcomes could not be assessed in this study.”

  1. The number of subjects at each follow-up visit, how follow-up data were taken it is not cleat nor is stated anywhere in the paper. Please clarify.

Reply:

Thank you for the comments. In this study, we enrolled patients with acute ischemic stroke between July 1, 2016, and December 31, 2017. All of these patients received a clinical follow-up schedule that was arranged every 3 months after discharge. Those patients who were lost to follow-up during scheduled clinical visits, refused blood serum tests, had missing baseline eGFR or creatinine data, had less than 2 sets of follow-up renal function data, or had a diagnosis of stroke mimics were excluded from this study. This information was described in section 2.1. Study design and dataset. However, to further clarify the follow-up conditions and follow-up visits of the whole study cohort, we added information to the Results section: “In this study, data from a total of 957 patients were retrospectively collected, and a total of 637 patients were enrolled in the analysis according to the above exclusion criteria (Figure 1). The mean follow-up duration was 26.9 ± 10.2 months, and the average number of follow-ups was 9 ± 3.4 visits.” Thank you for your very helpful comments.

  1. More than complicated logistic regression analysis continuous one are very important. Dichotomization always means lost of information. Please provide also data on correlation and linear regression model of GFR with ABI and baPWV.

Reply:

Thank you for the comments. In this study, eGFR was longitudinal, repeated-measure continuous data, and a linear mixed-effect model may be an appropriate tool to analyze the correlation between ABI/baPWV and eGFR while expressing eGFR as continuous data. We performed a mixed effect model to analyze the independent association between eGFR and ABI/baPWV. The tables are presented in the following section. We found that poststroke patients with abnormal ABI still had poor poststroke eGFR changes after adjusting for potential confounding factors, including age, sex, comorbidities and poor discharge mRS. In addition, older age, higher BMI, and HTN were also independently correlated with poor eGFR during follow-up. While we changed the factor of abnormal ABI from a dichotomized variable (indicating to separate ABI to lower than 0.9 and higher than 0.9) to a continuous variable (indicating the raw data of ABI), we found that higher ABI may be correlated with better eGFR data during follow-up. However, baPWV did not show a significant correlation with poststroke eGFR data. These data were in line with our findings regarding ABI/baPWV and different renal outcomes in the manuscript. Therefore, we did not put these data into our formal manuscript. Thank you for your very helpful comments.

(7.1) Mixed effect model of eGFR and abnormal ABI

Mixed effect model

Variables

b±SE

P value

Age

-0.39±0.07

<0.001

Male

-0.16±1.81

0.931

BMI

-0.50±0.22

0.023

Abnormal ABI (£0.90) vs. Normal ABI (>0.90)

-5.54±2.34

0.018

Hyperlipidemia

2.03±1.77

0.250

Heart disease

-3.98±2.31

0.086

Hypertension

-3.68±1.87

0.0495

Diabetes Mellitus

-2.40±1.85

0.195

Poor discharge mRS (³2)

-0.71±1.88

0.707

Adjusting factors: age, sex, BMI, ABI, comorbidities, and poor discharge mRS

(7.2 )Mixed effect model of eGFR and ABI (continuous variable)

Mixed effect model

Variables

b±SE

P value

Age

-0.39±0.07

<0.001

Male

-0.56±1.80

0.757

BMI

-0.51±0.22

0.019

ABI

13.93±5.83

0.017

Hyperlipidemia

1.94±1.77

0.272

Heart disease

-3.61±2.33

0.122

Hypertension

-3.79±1.87

0.043

Diabetes Mellitus

-2.11±1.86

0.256

Poor discharge mRS (³2)

-0.42±1.90

0.825

Adjusting factors: age, sex, BMI, ABI, comorbidities, and poor discharge mRS

(7.3) Mixed effect model of eGFR and abnormal baPWV

Mixed effect model

Variables

b±SE

P value

Age

-0.42±0.07

<0.001

Male

-0.68±1.87

0.717

BMI

-0.48±0.22

0.030

baPWV ³1.4 vs. baPWV <1.4

1.71±3.67

0.641

Hyperlipidemia

1.60±1.82

0.380

Heart disease

-4.14±2.41

0.087

Hypertension

-4.23±1.94

0.030

Diabetes Mellitus

-2.10±1.91

0.273

Poor discharge mRS (³2)

-0.69±1.92

0.720

Adjusting factors: age, sex, BMI, baPWV, comorbidities, and poor discharge mRS

(7.4) Mixed effect model of eGFR and baPWV (continuous variable)

Mixed effect model

Variables

b±SE

P value

Age

-0.39±0.07

<0.001

Male

-0.64±1.86

0.733

BMI

-0.48±0.22

0.033

baPWV

-0.16±0.12

0.200

Hyperlipidemia

1.49±1.82

0.411

Heart disease

-4.14±2.41

0.086

Hypertension

-3.91±1.92

0.043

Diabetes Mellitus

-1.71±1.92

0.374

Poor discharge mRS (³2)

-0.71±1.91

0.710

Adjusting factors: age, sex, BMI, baPWV, comorbidities, and poor discharge mRS

  1. Regarding ABI one important paper related to its use in cardiovascular prevention could be cited (10.1016/j.atherosclerosis.2020.11.004.)

Reply:

Thank you for the helpful comments. We added this reference as one of our references while introducing ABI (Reference 13). Thank you for your helpful comments.